The spleen bacteriome of wild rodents and shrews from Marigat, Baringo County, Kenya

Liyai Rehema 1 2
Kimita Gathii 2
Masakhwe Clement 2
Abuom David 3
Mutai Beth 2
Onyango David Miruka 1
Waitumbi John john.waitumbi@usamru-k.org 2
1 Department of Zoology, Maseno University College , Kisumu , Kenya
2 Basic Science Laboratory, United States Army Medical Research Directorate-Africa , Kisumu , Kenya
3 Entomology Section, United States Medical Research Directorate-Africa , Kisumu , Kenya
Edwards Scott
Electronic publication date: 2021 Sep 2
Publication date: 2021
Volume: 9
Electronic Location ID: e12067
Received 2020 Dec 13; Accepted 2021 Aug 5
Copyright: ©2021 Liyai et al.
Copyright year: 2021
Copyright holder: Liyai et al.
License: This is an open access article, free of all copyright, made available under the Creative Commons Public Domain Dedication. This work may be freely reproduced, distributed, transmitted, modified, built upon, or otherwise used by anyone for any lawful purpose.
License URL: https://creativecommons.org/publicdomain/zero/1.0/

Keywords: Wild mammals, Rodents, Shrews, Zoonosis, Spleen microbiome, 16S rRNA, Bacteria diversity, Next Generation Sequencing

Funding: Armed Forces Health Surveillance Division, Global Emerging Infections Surveillance (GEIS) Branch ProMIS ID 17_KY_1.3.8 (2017/2019) This work was supported by the Armed Forces Health Surveillance Division, Global Emerging Infections Surveillance (GEIS) Branch, ProMIS ID 17_KY_1.3.8 (2017/2019). The funders had no role in study design, data collection and analysis, decision to publish, or preparation of the manuscript.

==============================
Background

There is a global increase in reports of emerging diseases, some of which have emerged as spillover events from wild animals. The spleen is a major phagocytic organ and can therefore be probed for systemic microbiome. This study assessed bacterial diversity in the spleen of wild caught small mammals so as to evaluate their utility as surveillance tools for monitoring bacteria in an ecosystem shared with humans.

Methods

Fifty-four small mammals (rodents and shrews) were trapped from different sites in Marigat, Baringo County, Kenya. To characterize their bacteriome, DNA was extracted from their spleens and the V3–V4 regions of the 16S rRNA amplified and then sequenced on Illumina MiSeq. A non-target control sample was used to track laboratory contaminants. Sequence data was analyzed with Mothur v1.35, and taxomy determined using the SILVA database. The Shannon diversity index was used to estimate bacterial diversity in each animal and then aggregated to genus level before computing the means. Animal species within the rodents and shrews were identified by amplification of mitochondrial cytochrome b (cytb) gene followed by Sanger sequencing. CLC workbench was used to assemble the cytb gene sequences, after which their phylogenetic placements were determined by querying them against the GenBank nucleotide database.

Results

cytb gene sequences were generated for 49/54 mammalian samples: 38 rodents (Rodentia) and 11 shrews (Eulipotyphyla). Within the order Rodentia, 21 Acomys, eight Mastomys, six Arvicanthis and three Rattus were identified. In the order Eulipotyphyla, 11 Crucidura were identified. Bacteria characterization revealed 17 phyla that grouped into 182 genera. Of the phyla, Proteobacteria was the most abundant (67.9%). Other phyla included Actinobacteria (16.5%), Firmicutes (5.5%), Chlamydiae (3.8%), Chloroflexi (2.6%) and Bacteroidetes (1.3%) among others. Of the potentially pathogenic bacteria, Bartonella was the most abundant (45.6%), followed by Anaplasma (8.0%), Methylobacterium (3.5%), Delftia (3.8%), Coxiella (2.6%), Bradyrhizobium (1.6%) and Acinetobacter (1.1%). Other less abundant (<1%) and potentially pathogenic included Ehrlichia, Rickettsia, Leptospira, Borrelia, Brucella, Chlamydia and Streptococcus. By Shannon diversity index, Acomys spleens carried more diverse bacteria (mean Shannon diversity index of 2.86, p = 0.008) compared to 1.77 for Crocidura, 1.44 for Rattus, 1.40 for Arvicathis and 0.60 for Mastomys.

Conclusion

This study examined systemic bacteria that are filtered by the spleen and the findings underscore the utility of 16S rRNA deep sequencing in characterizing complex microbiota that are potentially relevant to one health issues. An inherent problem with the V3-V4 region of 16S rRNA is the inability to classify bacteria reliably beyond the genera. Future studies should utilize the newer long read methods of 16S rRNA analysis that can delimit the species composition.

Introduction

Infectious diseases remain the leading cause of death and morbidity worldwide. Of these, more than half are zoonoses that use domestic and wild animals as reservoirs (Galan et al., 2016). Rodents in particular play a major role in infectious disease ecology and epidemiology (Meerburg, Singleton & Kijlstra, 2009) the most famous being plague that dates as far back as 6th and 7th centuries (Hays, 2005). The fact that they occupy a vast range of habitats that interface between wildlife and urban communities make them ideal agents for carrying pathogens to humans and domestic animals. Given the accelerated anthropogenic encroachment to wildlife habitats, opportunities for contracting zoonoses will undoubtedly increase (Diagne et al., 2017; Wilson & Reeder, 2005). Thus, broad microbiome surveillance in rodents may help in understanding the diseases risk they pose, be they endemic or emerging zoonoses and in the end, formulating strategies to limit their spread (Vayssier-Taussat et al., 2014).

Small mammals which represent 40% of mammalian species are distributed in all continents except Antarctica (Delić et al., 2013) and they have benefited from human movement, enabling them to spread worldwide. The list of small mammals is large and among others, it includes rats, mice, squirrels, porcupines, beavers, guinea pigs and hamsters. Some of the small mammals are kept as pets (squirrels and hamsters), research animals (guinea pigs), or food animals (Gruber, 2016). Small animals harbor disease causing microbes that include a vast range of bacteria, protozoa, viruses and helminths that can be transmitted directly or indirectly to humans (Lindahl & Grace, 2015). Rats and mice have been reported to spread over 35 diseases worldwide which includes Hantavirus pulmonary syndrome, Hantavirus hemorrhagic fever and renal syndrome, plague, leptospirosis, salmonellosis, brucellosis, rat-bite fever, tularemia and Lassa fever (CDC, 2016; CDC, 2018), among others. Such diseases may be transmitted through inhalation or direct contact with fomites contaminated with feces, urine or saliva, handling of rodents and rodent bites. Other diseases are transmitted through vector intermediates such as ticks, lice, fleas, and mites. The increasing incidences of emerging, re-emerging and novel pathogens have led to renewed interest in the microorganisms carried by rodents (Schmidt et al., 2014).

Next generation sequencing (NGS) is particularly suited for studying microbial community in body organs. It allows unbiased detection of sequences that can be queried against genome databases to reveal the spectrum of pathogens represented in the sequences (Motro & Moran-Gilad, 2017). This is unlike targeted approaches that require some expectation of the diagnosis and works well for the detection of known endemic pathogens. An approach that combines targeted amplification and unbiased NGS has a major advantage of bringing the power of PCR amplification and the sequencing depth of NGS. The 16S rRNA has specific sites that are conserved in bacteria and can thus be amplified by PCR and the amplicons sequenced by NGS to identify bacteria biota in a sample (Couper & Swei, 2018; Motro & Moran-Gilad, 2017).

There is a specific association between microbes and their host reservoirs, and accurate taxonomic assignment of rodents/shrews is essential for a better understanding of the probable occurrences of the pathogens they carry (Lu et al., 2012; Müller et al., 2013). Morphological classification that relies on structural differences has been found wanting and miss-classification to species level occurs due to similar indistinguishable characteristics. Although rodents used in the current study are not difficult to distinguish from each other, the morphological classification performed initially (Masakhwe et al., 2018) had failed to speciate 14 shrews in the genus Crocidura. We therefore used molecular methods on all the samples to allow uniform taxonomic approach and to allow speciation of the unconfirmed host species. Molecular biology has revolutionized taxonomy and provides a higher resolution and has thus been proven to be better in unraveling hidden mammalian species that are over looked when morphological methods alone are used (Lu et al., 2012). Mitochondria genes such as the cytochrome c oxidase subunit 1 (cox1 or CO1) and cytochrome b (cytb) gene are commonly used for mammalian bar-coding (Pentinsaari et al., 2016; Tobe, Kitchener & Linacre, 2009). The cytb gene is about 1,149 base pairs and it offers more taxonomic information compared to cox1 (Tobe, Kitchener & Linacre, 2009) and therefore gives more accurate reconstructions and better resolution for separating species (Nicolas et al., 2012).

This study used deep sequencing to explore the identity and bacterial diversity in wild caught small mammals. We highlight the occurrence of potentially pathogenic tick borne zoonotic bacteria in some of the animals, and show differences in bacterial carriage across the small mammals.

Materials & Methods

Sample acquisition

The animal use protocol for this study was reviewed and approved by Walter Reed Army Institute of Research under protocol number AP-12-001, KEMRI IACUC #2208, and National Museums of Kenya NMK/SCom2013/08. Archived spleen tissues from a previous study were used (Masakhwe et al., 2018). Briefly, the rodent survey was conducted between 8th and 14th of June 2017. Each night, Sherman collapsible rodent traps were set up in a variety of habitats including cropped fields, grass fields, gardens, orchards, and around buildings. Trapped animals were removed and after euthanasia, the surface of the rodent abdomen was cleaned with 70% ethanol before necropsy. About 100 mg of the spleen samples were collected, stored in screw cap plastic vials before placing them in shippers containing liquid nitrogen. The tissue biopsies were later transferred to the laboratory where they were stored at −80 °C.

Genomic DNA isolation from the spleen tissues

Total genomic DNA was isolated from ∼10 grams of spleen tissues using QIAamp DNA Mini Kit (Qiagen, Valencia, CA, USA) as recommended by the manufacturer. DNA was eluted in 100 µL and stored in a −80 °C freezer until required.

Molecular taxonomy to aid identification of individual animal species

The primers used to amplify the cytb gene are listed in Table S1. Where amplification could not be achieved with the primary PCR primers, a secondary nested PCR was performed using internal primers L14749 and H14896 as described by (Nicolas et al., 2012). PCR was performed in a 25 µL reaction volume containing 2 µL of DNA template, 0.2 µM of each primer, 0.05 U of MyTaq polymerase and 5 µL of 5X MyTaq buffer (Bioline, UK). Amplification was performed in an Eppendorf Master cycler pro 384 (Eppendorf, USA) with an initial denaturation step of 94 °C for 2 min, 30 cycles of 94 °C for 30 s, 52 °C for 30 s, and 72 °C for 1 min followed by a final extension step at 72 °C for 10 min. Amplicons were visualized on a 2% agarose gel (Thermo Fisher Scientific, CA, USA) stained with GelRed (Biotium, Australia). PCR products from positive samples were purified using AMPure XP beads (Beckman Coulter, CA, USA). The forward and reverse primers were used for cycle sequencing by the Big Dye Terminator Cycle Sequencing Kit v 3.1 (Applied Biosystems, CA, USA). Reaction products were then purified using CleanSEQ beads (Beckman Coulter, CA, USA) and sequenced on a 3130 Genetic Analyzer (Applied Biosystems CA, USA).

Spleen bacterial diversity by 16S rRNA deep sequencing

The bacteria DNA in the spleen biopsy was amplified with primers that target the 16S rRNA V3–V4 hyper variable region (Table S1). The primers were tagged with Illumina sequencing adapters as previously described (Klindworth et al., 2013). A non-target control sample (PCR water) was used to track laboratory contaminants. Amplification was carried out in a 25 µL reaction consisting of 12.5 µL of 2X NEB Next PCR master mix (New England Biolabs, MA, USA), 0.2 µM of each primer and 2.5 µL of DNA template. Amplification was performed on the Eppendorf Master cycler pro 384 (Eppendorf, Humberg, Germany) with an initial denaturation at 95 °C for 3 min followed by 25 cycles of 95 °C for 30 s, 55 °C for 30 s, 72 °C for 30 s and a final extension of 72 °C for 5 min. Amplicons were thereafter purified using AMPure XP beads according to the manufacturer’s instructions (Beckman Coulter Genomics, Brea, CA, USA).

A dual indexing PCR to allow multiplexing of samples was done in a 50 µL reaction consisting of 5  µL of purified amplicons, 5  µL of each Nextera XT i7 and i5 Index Primer (Illumina, USA), 25 µL of NEBNext High-Fidelity 2X PCR Master Mix (New Englands BioLabs, MA, US) and 10 µL of PCR grade water (Thermo Fisher Scientific, Waltham, MA, USA), with thermocycling at 95 °C for 3 min, followed by 12 cycles of 95 °C for 30 sec, 55 °C for 30 sec, and 72 °C for 30 sec, and a final extension at 72 °C for 5 min. The indexed amplicon libraries were purified with AMPure XP beads according to the manufacturer’s instructions (Beckman Coulter Genomics, Brea, CA, USA), and then quantified on Qubit Fluorometer 2.0 using Qubit dsDNA HS assay kit according to the manufacturer’s protocol (Thermo Fisher Scientific, Waltham, MA, USA). Libraries were normalized to a concentration of 4 nM and then pooled. The pooled samples were denatured and diluted to a final concentration of 12 picomoles, then spiked with 5 % PhiX (Illumina, USA) as a sequencing control. Samples were sequenced on MiSeq platform (Illumina, USA) using MiSeq 600 cycle reagent kit V3 (Illumina, USA).

Data analysis

Forward and reverse nucleotide sequences from the cytb gene were quality checked and assembled into contigs using CLC main work bench version 7.5 (CLC Inc, Aarhus, Denmark). Identification of species within the orders Rodentia and Eulipotyphyla was done by querying the assembled contigs against the nucleotide database (GenBank) using the Nucleotide Basic Local Alignment Search Tool (BLASTn) (Altschul et al., 1990).

To determine the phylogenetic placements of the small mammals, reference sequences corresponding to the four small mammal species i.e., Acomys wilsoni, Mastomys spp, Crocidura spp and Rattus rattus were downloaded from GenBank (http://www.ncbi.nlm.nih.gov/genbank/). Multiple sequence alignments were performed in MEGA version 7 (Kumar, Stecher & Tamura, 2016) and the General Time Reversal model with gamma distribution and invariable sites (GTR+ Γ+I) was determined as best model for the analysis. The alignments were then used to infer phylogenetic relationship using the maximum likelihood treeing method. Branch support was calculated as 1,000 bootstrap replicates.

For the 16S rRNA deep sequencing, the Miseq sequences output were de-multiplexed and adapters trimmed using the Miseq reporter software version 2.6.3 (Illumina, USA). Mothur version 1.35 pipeline was used for paired end reads contig assembly, sequence quality filtering, chimera removal and taxonomic identification (Schloss et al., 2009). In brief, contigs containing ambiguous bases, and those with lengths ≤389 bp and ≥500 bp were discarded. The SILVA database was customized for the V3-V4 region and used for sequence alignment, followed by merging sequences that were not more than 2 bp different from each other using the pre-cluster command in Mothur (Quast et al., 2013). Unassigned OTUs and those assigned to Chloroplast, Mitochondria, Archaea, and Eukaryota were discarded. Samples with less than 1,000 (n = 10) sequences were discarded as small library sizes are often a confounding factor that obscures biologically meaningful results (Weiss et al., 2017). Taxa contained in the non-template control were censored from sample dataset as proposed by Davis et al. (2018).

Statistical analysis and visualization were performed using R software environment (version 4.1, R Core Team, 2016), with the following add-on statistical packages: ‘phyloseq’ (McMurdie & Holmes, 2013), ‘vegan’ and ‘ggplot2’ (Oksanen et al., 2014; Wickham, 2009). Rarefaction was performed to down sample the data for alpha diversity analysis using the rarefy_even_depth command in phyloseq with replacement. This was done to account for unequal sequencing between samples (Weiss et al., 2015). The rarefied data was used to compute the Shannon diversity index (Shannon, 1948), by first determining the bacterial diversity for each animal, and then aggregated to genus level before computing the mean.

Results

Small mammal speciation by cytb gene

A total of 54 spleen tissues from wild caught small mammals were used for molecular taxonomy and microbiome profiling. The small mammals had previously been classified based on morphological characteristics into two major orders, Rodentia and Eulipotyphyla (Masakhwe et al., 2018). The order Rodentia (38/54 =70.4%) had the following species: Acomys wilsoni (n = 21), Lophuromys sikapusi (n = 1), Mastomys natalensis (n = 8), Rattus rattus (n = 2), Arvicanthis niloticus (n = 6). The order Eulipotyphyla (16/54 =29.6%) had Crocidura olivieri (n = 2), and other un-speciatable Crocidura spp (n = 14).

Using cytb gene, 49/54 samples generated usable sequences. Homology searches with BLASTn against the GenBank database classified the small mammals into two main orders: Rodentia (77.6%) comprising Acomys (21/49), Mastomys (8/49), Arvicanthis (6/49) and Rattus (3/49), and Eulipotyphyla (22.4%), comprising only Crocidura (11/49).

Of the 49 samples sequenced, only 46 could be used for phylogenetic analysis. These were compared against 20 representative species in a maximum likelihood tree. As shown in Fig. 1, the shrews (11/46) clustered with Crocidura somalica in a well-supported clade (99%). The rodent species were more diverse. 20/46 samples branched with Acomys wilsoni (100%), 3/46 samples branched with Rattus rattus, 6/46 branched with Arvicanthis niloticus and 6/46 branched with Mastomys natalensis.

Figure 1 Phylogenetic tree showing samples from this study against reference species.

The tree was inferred from the cytb gene alignment using Maximum likelihood methods. 46 study samples (all have prefix “R” and 20 reference isolates from genbank marked with an *) were used to infer the Maximum Likelihood tree. Blue lines represent members of orderRodentia, black lines represent Eulipotyphla. Numbers on the branches represent bootstrap support values >50%. The branch lengths represent the number of base substitutions per site.

Diversity of bacteria in the small mammals

Sequences from the 54 spleen samples yielded 8,621,961 raw sequence contigs. After quality filtering, collapsing duplicate sequences, removing chimeras and non-bacterial sequences, 2,778,822 sequences were considered suitable for further analysis. On querying the SILVA rRNA database, the sequences were grouped into 182 taxa at 97% sequence similarity. Of the 54 spleen samples examined, 10 that had small library sizes (<1000 sequences) were dropped from downstream analysis. The remaining 44 samples had a total of 2,778,038 sequences. After contamination screening a total of 7 OTUs were identified as contaminants by the prevalence method in the “decontam” R package (see Table S2 for list of contaminants) and were removed from the spleen sample dataset.

Figure 2 shows the 17 bacteria phyla that were detected. Proteobacteria was the most abundant and contributed 35,122 of 51,731 total contigs (67.9%). Other phyla included Actinobacteria at 16.5%%, Firmicutes (5.5%), Chlamydiae (3.8%), Chloroflexi (2.6%) and Bacteroidetes (1.3%). Less abundant phyla (<1%) included Acidobacteria, Verrucomicroba, Planctomycetes, Fusobacteria, Deinococcus-Thermus, Armatimonadetes, Gemmatimonadetes, TM7, OD1, Spirochaetes and SR1.

Figure 2 Bar plot showing bacteria phyla in the spleen samples from small animals.

Of the 17 bacteria phyla identified, Proteobacteria was the most abundant, followed by Actinobacteria except in Arvicanthis where Farmicutes were second dominant.

Taxonomic assignment at phylum and genus level for the 182 OTUs is shown in Fig. 3. Of these genera, Bartonella was the most abundant with sequence yield of 45.6% (19,872 out of 43,622 total contigs). Because some of the bacteria could not be classified beyond family level, the total contigs at genera level were fewer (43,622) than those at phylum level (51,731 contigs). Compared to Bartonella, the relative abundance of other bacteria in the Proteobacteria was low: Anaplasma (8.0%), Delftia (3.8%), Methylobacterium (3.5%), Coxiella 2.6%, Bradyrhizobium (1.6%) and Achromobacter (1.2%), Acinetobacter (1.1%). Potentially pathogenic bacteria from other phyla occurred at <1% and included Ehrlichia, Rickettsia and Brucella.

Figure 3 Circular bar plot showing taxonomic assignment for the 182 OTUs in the spleen samples from small animals.

Of the bacteria genera identified, Bartonella was the most abundant and contributed 45.6% of the total contigs.

Figure 4 shows a heat map of 44 spleen samples from the different small mammal species clustered against the bacteria genera with abundance >5%. Of the 44, 34 had Bartonella sequences. Of these, 20 (7 from Mastomys, 5 Crucidura, 4 Arvicanthus, 3 Acomys and 1 Ratus) contributed to the over-representation of Bartonella sequences in the heat map. Interestingly, the spleen samples with over-representation of Bartonella tended to have fewer or no other bacteria genera.

Figure 4 Heatmap depicting the differential abundance of microbial taxa among the samples.

The Y-axis represent the samples from the different small mammal species. The X-axis represent microbial taxa at the genus level and are ordered by hierarchical clustering. The color scale represents the scaled abundance of each bacteria with red indicating high abundance and green indicating low abundance. Sequence abundance was highest for Bartonella with 20 samples (seven from Mastomys, five Crucidura, four Arvicanthus, three Acomys and one Ratus) contributing to over-representation of this genus.

The Shannon diversity index ranged from 0.3204 to 3.5135, indicating a broad variation in the bacterial diversity between samples. The mean bacteria diversity in Acomys was 2.86 compared to 1.77 for Crocidura, 1.44 for Rattus, 1.40 for Arvicathis and 0.60 for Mastomys (Fig. 5). Using one-way ANOVA followed by the Tukey HSD (Honest Significant Difference) posttest, the bacteria diversity was significantly higher in Acomys compared to Mastomys (p = 0.016).

Figure 5 Boxplot showing median bacterial diversity across the small mammal genera as measured by Shannon diversity index.

Microbiota in Acomys spleen was more diverse than in other small mammals, but only significantly higher than Mastomys (p = 0.016).

Discussion

In the present study, 54 wild caught small mammals that were previously classified using phenotypic characters were re-identified by molecular barcoding using cytb gene. As shown in Fig. 1, the small mammals clustered in five clades: Acomys, Mastomys, Arvicanthis, Rattus and Crocidura. Unfortunately, and unlike morphological classification, not all the 54 mammalian samples were classified by Ctyb as we could not obtain complete or good sequences in some of the samples. The 46/54 samples that had good quality sequence were identified to species level, supporting earlier claims on utility of the cytb gene in mammalian classification (Nicolas et al., 2012; Pentinsaari et al., 2016; Tobe, Kitchener & Linacre, 2009). In the current study, cytb was able to resolve mis-identification that was reported in a previous study that had used morphological markers (Masakhwe et al., 2018). In that study, all the 11 shrews (order Eulipotyphyla) could not be classified beyond the genus Crocidura. By cytb barcoding, all the 11 branched with the C. somalica (Fig. 1). Other discrepancies included rodent R003 that was morphologically identified as a Lophuromys spp, but cytb barcoding identified it as Arvicanthis niloticus. As shown on the phylogeny tree, this sample branched with other Arvicanthis group in a clade that had with high branch support (100% Bootstrap support values).

16S rRNA has previously been used to study spleen microbiome in wild small mammals (Ge et al., 2018; Razzauti et al., 2015) but to the best of our knowledge, ours is the first study to evaluate spleen microbial diversity in wild caught small mammals in Kenya. A total of 17 phyla (Fig. 2) that contained 182 bacteria genera (Fig. 3) were detected. Unlike in the study by Ge et al. where Firmicutes dominated, followed by Proteobacteria (Ge et al., 2018), in our study, Proteobacteria, a major phylum of Gram-negative bacteria was the most dominant, accounting for 67.9% of the spleen microbiota. The phylum has a wide variety of pathogenic genera and others that are nonparasitic free-living. Actinobacteria, a Gram-positive phylum was the second dominant at 16.5%. Other phyla included Firmicutes at 5.5%, Chlamydiae (3.8%), Chloroflexi (2.6%) and Bacteroidetes (1.3%). Other less abundant phyla (<1% sequence abundance) included Acidobacteria, Verrumicroba, Planctomycetes, Fusobacteria, Deinococcus-Thermus, Armatimonadetes, Gemmatimonadetes, TM7 (Saccharibacteria), OD1 (Parcubacteria), Spirochaetes and SR1 (Absconditabacteria).

Of the 182 bacteria genera, Bartonella was the most abundant (45.6%) (Fig. 3). This is similar to a study in wild voles in France that found Bartonella as the dominant zoonotic genus in the spleen (Tołkacz et al., 2018). Other genera carrying potentially infectious pathogens that were identified in our study include Anaplasma (8.0%), Coxiella (2.6%), Acinetobacter (1.1%). Ehrlichia, Rickettsia, Brucella, Capnocytophaga, Mycobacterium and Borellia occurred at <1% abundance. Similar pathogen carriage by rodents was reported previously (Ge et al., 2018; Han et al., 2015). A number of the pathogens identified are tick borne zoonoses, including Anaplasma, Coxiella, Ehrlichia, Rickettsia and Borellia, reinforcing the reservoir role played by wild caught small mammals in maintenance and transmission of these zoonotic pathogens. One important shortcoming of using the V3-V4 region of the 16S rRNA in microbial analysis is its inability to resolve bacteria genera up to the species level. Therefore, for complex genera that have both pathogenic and non-pathogenic species (eg Coxiella), it becomes difficult to call these pathogens with confidence (Bonnet et al., 2017).

A previous study (Masakhwe et al., 2018), had identified Orientia in chigger mites collected from the same small mammals that were used as the source of spleen samples in the current study. Our study did not identify Orentia in the spleens. This may be due to the fact that mites are the vectors and amplify the bacteria and hence are more abundant in the vector. A previous spleen microbiota study in France had identified Orentia that was divergent from O. Tsutsugamushi (Razzauti et al., 2015). Like in the study reported by Razzauti et al. (Razzauti et al., 2015), the sequence of the Kenyan Orentia was closer to O. chuto than O. Tsutsugamushi (Masakhwe et al., 2018).

As shown in Fig. 4, of the 44 spleen samples analyzed, 34 (77.3%) contained Bartonella sequences. Twenty spleen samples (7 from Mastomys, 5 Crucidura, 4 Arvicanthus, 3 Acomys and 1 Ratus) contributed to the over-representation of Bartonella sequences. No other bacteria genus had such dominance. Interestingly, the spleen samples with over-representation of Bartonella had fewer or no other bacteria genera. It is tempting to attribute this dominance to competitive exclusion, a phenomenon that is well established in the intestinal tract of food animals (Callaway & Martin, 2006; CaritaSchneitz, 2005; Fuller, 1989). The aggregated mean Shannon diversity index in the Acomys was 2.86, higher than all other small animals, but only statistically higher than Mastomys (p = 0.016) (Fig. 5). It is worth noting that the statistics suffer from lack of adequate sample size in some animal species and therefore the calculated mean Shannon diversity should be interpreted with caution.

An inherent problem with the V3-V4 region of 16S rRNA gene that was used in this study is that it is only ∼460 bp. Although adequate for determining the genera, it is not polymorphic enough to resolve the bacteria to species level. As we went to press, a new method that amplifies full-length 16S rRNA genes was described and was shown to be capable of identifying and classifying bacteria to species and strain level (Callahan et al., 2020). Going forward, further characterization using such methods will be required not only to discriminate between species and strains, but to also identify pathogenic and non-pathogenic (endosymbionts) species in complex genera such as Coxiella and Rickettsia.

Conclusions

The small mammals that contributed to the bacteria diversity data came from a semi-arid ecosystem in northern Rift Valley, Kenya (Masakhwe et al., 2018). The current study extended this work by analyzing the bacterial abundance and diversity in the spleen of these wild caught small mammals. Among the 182 bacteria genera detected, four tick borne zoonotic bacteria, namely Anaplasma, Ehrlichia, Borrelia, and Rickettsia, and others that are shed in the environment by the livestock (Coxiella, Brucella and Leptospira) were identified(Rees et al., 2021). These findings illustrate the utility of wild animals as surveillance tools for monitoring microbiome that are potentially relevant to one health issues. Although the spleen is an ideal organ to investigate systemic microbiome, organs such as the digestive system (feces) would add useful information on environmental contamination. Going forward, such studies should, in addition to using full length 16S rRNA, incorporate viral pathogen discovery in order to comprehensively monitor for introduction of novel pathogens with potential for zoonotic spillover.

Disclaimer

Material has been reviewed by the Walter Reed Army Institute of Research. There is no objection to its publication. The opinions or assertions contained herein are the private views of the authors, and they are not to be construed as official, or as reflecting true views of the Department of the Army or the Department of Defense.

Supplemental Information

Supplemental Information 1 Primers used for amplification and sequencing of Cyt b and 16S rRNA genes

Note: Under the primer column, the letters “L” and “H” identify the light (L) and heavy (H) chains respectively and the number gives the position of the 3′ base of the primer. # = primers used for nested PCR.

Click here for additional data file.

Supplemental Information 2 Number of reads in non-template control sample that was used to track contaminant OTUs

Click here for additional data file.

Additional Information and Declarations

Competing Interests

Author Contributions

Animal Ethics

Field Study Permissions

Data Availability

The authors declare there are no competing interests.

Rehema Liyai and Gathii Kimita performed the experiments, analyzed the data, prepared figures and/or tables, and approved the final draft.

Clement Masakhwe performed the experiments, authored or reviewed drafts of the paper, and approved the final draft.

David Abuom performed the experiments, authored or reviewed drafts of the paper, assisted in rodent trapping, morphological identification and necropsy, and approved the final draft.

Beth Mutai performed the experiments, analyzed the data, authored or reviewed drafts of the paper, and approved the final draft.

David Miruka Onyango analyzed the data, authored or reviewed drafts of the paper, and approved the final draft.

John Waitumbi conceived and designed the experiments, authored or reviewed drafts of the paper, and approved the final draft.

The following information was supplied relating to ethical approvals (i.e., approving body and any reference numbers):

The animal use protocol for this study was reviewed and approved by Walter Reed Army Institute of Research under protocol number AP-12-001, KEMRI IACUC #2208, and National Museums of Kenya NMK/SCom2013/08.

The following information was supplied relating to field study approvals (i.e., approving body and any reference numbers):

Field experiments were approved by the Kenya Medical Research Institute as part of the IACUC approval (#2208).

The following information was supplied regarding data availability:

The raw reads of the 16S metagenomics are available at NCBI BioProject ID: PRJNA669316.

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
