# Peer review of "The spleen bacteriome of wild rodents and shrews from Marigat, Baringo County, Kenya"

_PeerJ, doi:10.7717/peerj.12067_

## Round 0.1 · original submission · Major Revisions

Reviewer 1 has a number of useful comments to improve the manuscript. In particular, this reviewer suggests stating specific hypotheses and quantifying and testing for bacteriome diversity with appropriate statistical tests.

Reviewer 2, however, has serious concerns about the paper, especially the way in which the rodents were identified with DNA data. Perhaps you meant to ID the rodents to individual species, rather than to tell the difference between a shrew and a rodent? As the Reviewer states, you do not need DNA data to distinguish a shrew from a rodent.

Reviewer 1 ·

Basic reporting

The text is written fairly clearly, however, there are a number of grammatical issues that are distracting and detract from the paper. Additionally, there are several improvements that could be made to the basic reporting of this paper.
Firstly, there is very little literature review or additional context provided. The first line of the background statement discusses EIDs, but the authors do not discuss EIDs in depth, nor how microbiome metabarcoding is useful for monitoring EIDs. More information on prior microbiome studies on small mammals and the usefulness of the spleen as a sampling tissue is needed. The current introduction discusses only surface-level topics about small mammal pathogens.
There are also no stated hypotheses, although the authors do calculate Shannon diversity and assess differences among different species in the discussion. There is clearly an opportunity to test for differences in diversity among the different species, but the authors make no attempt to do this. In addition to conducting this analysis, the authors should also discuss the meaning of high/low Shannon diversity in the context of EIDs, as well as more discussion about the specific bacterial groups detected.
There are a few issues with some of the figures:
Figure 1: The font is entirely unreadable
Figure 2: Please provide sample sizes for each small mammal group
Figure 3: This figure is easier to read than figure 1, but is still challenging to see the bacterial groups.
Figure 4: This would be better as an ordination. At the very least, the samples should be grouped and labeled by small mammal species. It looks like Bartonella was very common in about half of the samples. Are these individuals from the same group? Additionally, the diverging color scheme is inappropriate for these data.
Figure 5: Providing statistical information on this plot would improve interpretation (e.g. which means are significantly different?)

Experimental design

There was no specific research question aside from a goal to sequence and identify the spleen microbiome. This study could be improved by examining differences in bacterial composition among the different rodent species. Similarly, more detail about the samples is needed (despite the citation to the more in-depth previous study, further information about sampling dates, storage method, and species breakdown is needed).
It is not entirely clear what knowledge gap is being filled, as the introduction does not review metabarcoding studies in small mammals, nor does it discuss whether the species examined in this study have been investigated in this manner before.
The metabarcoding and bioinformatics methods are generally clear and well-described; however there is very little discussion of rarefaction methods and diversity calculations/analyses.

Validity of the findings

There are no statistical tests of diversity differences among species, yet the authors claim in the discussion that diversity in Acomys is higher than in other genera. A test for diversity differences should be conducted if they want to support their claims with the data. Additionally, some kind of ordination of bacterial communities among genera (plus rays for specific bacterial groups) would also be more informative than the results as they are currently presented. Again, there was no clear research question posed in the introduction, so it is impossible to connect the conclusions in the discussion with the original study goals. A better review of the literature, justification for the study, and meaningful research question would improve this paper. Finally, the conclusion section did not seem to reflect the actual data from the study. It could have been written without any of the information from the study (i.e. the finding that the spleen acts as a microbial filter and that the 16S rRNA region can be used to characterize microbiota are findings from Ge 2018). While this study replicates the Ge 2018 study, the authors do not provide a clear rationale for why their study furthers our understanding (aside from saying that the specific species from the study location were examined). However, if the authors are able to provide a more in-depth analysis, especially of bacterial diversity, in additional to details about the ecology of the small mammal species and why this matters for disease transmission, then this study would provide a meaningful extension to the Ge 2018 study.

Additional comments

The data and sequences from this study are important, as they document bacterial communities in a range of small mammal genera of disease concern (e.g. Rattus) in Kenya. However, the study needs far more context and rationale, including a clear research question.

Reviewer 2 ·

Basic reporting

There are many errors in the paper. I forged ahead and read it anyway. It seemed to have promise. Introduction to the methods of barcoding was extremely superficial. The authors mix up classification with identification of mammals. Several references are incomplete.

Experimental design

no comment

Validity of the findings

Any student who had a basic course in mammalogy can identify a shrew versus a rodent. There was a lot of wasted effort sequencing these mammals when someone could just look at them. Also since there were no specimens deposited in museums the findings cannot be replicated or ever checked. Did the authors just throw away the mammals after the materials were taken from them?

Additional comments

Line 56: Some species are spread around like Mus musculus and M. domesticus and Rattus norvegicus and R. rattus and a few Sciurus spp. plus some others. Nutria (Myocastor coypus) have been transplanted from Bolivia to various continents. Beavers (Castor canadensis) are running rampant in Argentina, but for the most part those are just a few species of rodents that are sufficiently wide-spectrum in their ecological niches that they can be moved around by humans and then thrive where they are introduced.
Line 66. Rats and mice may harbor organisms like viruses and bacteria and helminths that can cause disease in animals, they do not transmit the disease. They may act as reserviors of the various pathogens that cause disease in other hosts.
The reference for these statements is incomplete:
368 Lindahl, J. F., & Grace, D. 2015. The consequences of human actions on risks for infectious 369 diseases: a review. 1, 1–11.
Line 76 paragraph. The authors mean identification instead of classification. Classification means the process of delimitation, ordering, and ranking of taxa. Identification means to find out what the species of mouse is in hand.
Line 78. Morphological classification means nothing. What the authors mean is identification of rodents using morphological characters.
LIne 85 - Accurate reconstructions of what?
I think that the literature on the barcoding initiative should be consulted to write this section.
https://ibol.org/about/dna-barcoding/

Line 237: I see no reason to go to the trouble to create a phylogeny of species that are clearly very distantly related. It does not make logical sense to spend the time and resources to do this and there is nothing that can be derived from it.f
Line 240: Then why do you say that morphological methods for identification is poor in the introduction? Here you show that this is not the case.

Unfortunately this paper was ill conceived: Trying to id the mammals using molecules when a look at the skull by any competent mammalogist would give the ID instantly. It is NOT POSSIBLE to mix up the identification of a shrew with a rodent unless the person doing the ID had never seen one before.
I cannot say much about the species of bacteria that were found in the spleen. The work seems good in that regard.

---

## Round 0.2 · Minor Revisions

Although reviewer 3 indicated major revisions, his/her comments are, I think, fairly easily addressed. I encourage you to address the comments of both reviewers, in particular discussing appropriate controls, details of methods and how various bacterial strains are identified.

Reviewer 1 ·

Basic reporting

Basic reporting in the article has been improved. However, there are still two areas that need more clarification:

1. How Shannon Diversity was calculated. Was this done by first aggregating relative read abundance (or even prevalence?) to species level (i.e gamma diversity), and then performing the calculation? Or by calculating individual-level SD and taking the mean/median?

2. The new statements attempting to clarify the meaning of the SD do the opposite (more below). The authors should either remove these or reformulate.

Finally, again, there are no attempts to make *statistical* hypotheses, even though the authors attempt to compare SD among species without using any estimates of error.

Experimental design

This is not relevant, as the authors have stated that they do not have a research question. However, given that they attempt to compare bacterial diversity among small mammal species, there clearly is an underlying interest in making statistical comparisons.
Methods appear reproducible.

Validity of the findings

Underlying data is provided, figures (mostly) accurately depict the data and findings (see comments attached).
Conclusions are somewhat overstated in that the authors seem to imply that by identifying small mammals with high spleen microbial diversity, they are also identifying species/individuals with high health risks to humans. This is not necessarily the case.

Additional comments

I appreciate the efforts that the authors have made to improve the text. There are several sections where additional text has provided more clarity, and the figures are much more readable and accessible.

Below are line-by-line comments:
All line number references are based on the line numbers in the tracked-changes document.

L19-20 Does higher bacterial diversity correlate with increased human health hazards? This seems like a bit of a stretch for this paper, as it is very difficult to make any statements about the pathogenicity of detected bacterial groups, nor their likelihood of transmission to humans.

L23-28 This section could be shortened substantially. Details about sequencing can be simply listed in the main text.

L29-30 If you intend to keep it as a single Shannon value for each species (i.e gamma diversity for each species, as opposed to a mean +/- SE on an individual level), then you could clarify that this is a species-level Shannon index, rather than an individual-level Shannon measure.

L30 Avoid using ‘individual animal species’. There are both individuals and different species, so statements like this are confusing.

L53-55 Clarify that this is across the whole species (gamma diversity), rather than an average across individuals, if this is the case. I’m still not sure what you calculated, because there are individual-level estimates shown in Figure 5, but the rebuttal comments seem to indicate that calculations were performed only at the species level. If this is the case, then there needs to be a statement about the fact that Shannon diversity is biased low at small sample sizes and if there was a correction for that (e.g. rarefaction, using a jack-knife estimate etc.). I also have some issues with this because the sample sizes are very different depending on the species. It’s hard to say much about species-level diversity patterns when there are only two samples (e.g. Rattus), and thus, Shannon Diversity for these low-sampled species is likely underestimated.

One alternative is to calculate mean individual-level Shannon diversity using relative read abundances as an abundance metric (it looks like this was done for Figure 5). Another is to report the mean number of bacterial groups per individual (richness). The total number of bacterial groups per species could be reported, but this also has issues with small sample sizes.

As shown in Figure 5, you clearly have a response metric (Shannon Diversity at the individual level) with several replicates per group, thus giving you the ability to estimate a mean (or median) and associated variation. This would then allow you to test the null statistical hypothesis that there is no difference in mean individual-level diversity among species (e.g. using a 1-way ANOVA).

L70-78 The additional text here greatly improves the study context and rationale.

L34-37 These sentences are not necessary.

L109-120 While this is a fair argument, I am not sure it is totally relevant to this study because the species investigated are not too difficult to distinguish from one another (at least for rodents). However, it seems that in this study, the researchers did not have access to the skins/preserved specimens, so perhaps a more relevant point is that NGS methods allow for more streamlined/accurate use of tissues of an unconfirmed host species.

L122-124 “Disease risk” is a strong term, as the pathogenicity of these bacterial groups is not being tested in this study. Simply stating that bacterial diversity, and in particular, diversity/presence of certain bacterial groups was investigated is sufficient.

L231-232 It is unclear how Shannon diversity was calculated at the species level. Was this simply based on the number of OTUs detected across all individuals of a given small mammal species?

L323-325 This statement is confusing and does not explain why significance can be drawn without estimates of variation/uncertainty.

L396-398 This statement is confusing as readers most likely know what Shannon Diversity is. Additionally, if a particular species had a microbiome twice as diverse as another, this would not necessarily indicate higher infectiousness.

L404 If these papers report the new method, then the statement ‘as we went to press’ does not make sense, as the papers were published in 2009 and 2016.

L415 A citation is needed for this statement.

L416 This statement seems a bit strong. I do not think that this study can definitively make a connection to human health hazards, which are complex and depend on many things (e.g. environmental transmission mechanisms, exposure routes, immunity, pathogenicity of investigated bacterial groups), but rather that it shows a method to determine microbial diversity in these animals, many of which may have been uncharacterized previously, and identifies species that are capable of carrying certain bacterial groups of potential human health interest.
The focus on Shannon Diversity does not relate directly to the article's attempt to connect the findings with potentially pathogenic bacterial groups.

There is potential to discuss the two species investigated here that are well-known for being involved in zoonotic infections (Rattus and Mastomys).

There is also more potential for more discussion as to why the spleen is investigated, and why this is useful for monitoring (as opposed to feces, for example, which represents material that enters the environment and is likely to serve as a source of exposure).

Figure 2:
I am assuming that these bar charts are the result of aggregating all reads within species (i.e. using the mean/sum)?

Figure 5:
This figure is very helpful; however, it is inaccurate to show small sample sizes using boxplots. The text states that there are only 2 samples for Rattus, yet a boxplot is shown. Additionally, the figure legend is confusing: does ‘alpha diversity measure’ refer to Shannon Diversity or some other alpha metric?

Reviewer 3 ·

Basic reporting

o Suggested places to review the use of “clear, unambiguous, technically correct language”: Line 19: not clear how “systemic microbiome” might be defined; consider in particular the common definitions of “microbiome” as a collection of living microorganisms together with their host, and whether DNA sequences obtained from the spleen can be expected to represent such a community.

o Line 323: Contrary to what is written, the study by Ge et al. (2018) reported Firmicutes as bacterial phylum that predominated the samples (>70%) whereas Proteobacteria were a more minor component (~23%). Please correct and clarify how this differs from the results reported in the present study.

o Whereas the text emphasized comparisons of bacterial taxa at the phylum level, Figures 3 and 4 emphasized OTUs identified to genus. These genus-level comparisons appear more relevant (and interesting) to the research aims the authors posed. For example, whereas there are many Proteobacteria OTUs in the data, they are (by far) predominated by Bartonella. Bartonella is a known pathogen with a reasonably well understood natural history, as the authors review in the discussion (lines 334-346). The paper could be improved by emphasizing through the basic reporting these predominant features of the data at a finer grain, and better describing the known life histories of the taxa involved to underscore relevance to the main research question; the current emphasis on phylum-level comparisons and the large number of taxa with particularly low sequence relative read abundances can obscure these predominant data features and could introduce qualitative errors.

o Throughout: minor typos and changes in tense or plurality should be checked. It may be possible to address these through copyediting.

Experimental design

o Commendable steps: I appreciated the use of mitochondrial DNA barcoding methods to identify small mammals and evaluate results in a phylogenetic context; this is a strength of the paper.

o Suggested review of whether the “research question is well defined”: Line 21: It would help to specify how monitoring DNA sequences in the spleens of small mammals could conceivably serve as an indicator for human health at this point in the abstract; there are many ways that human-rodent interactions involve health considerations, but evidence relevant to public health requires the ability to identify a pathogenic strain of bacteria (not necessarily possible using 16S markers) and a plausible transmission pathway (not to be taken for granted even in the presence pathogens). Could such a monitoring program be said to provide more direct information about the potential health status or rodents, with indirect implications for human health?

o A description of the experimental controls would be helpful, including any sequencing results obtained from extraction blanks and PCR controls. Several the putatively pathogenic bacteria listed in this paper are common laboratory contaminants in microbiome studies (see, e.g., the report by Salter et al., 2014). Proteobacteria in particular are common laboratory contaminants, and when the template is based on a low biomass sample these contaminants can appear to be a relatively abundant constituent of the data. A comparison of the sample data with controls would help rule out laboratory contamination as a concern.

Validity of the findings

o Line 44: it would help to clarify even in the abstract how potential pathogenic bacteria were identified from the strains, and issue a reminder that taxonomic classifications may provide an indication of *possibly* pathogenic bacteria without implying that these sequences definitively represent pathogenic organisms. Bradyrhizobium, for example, includes strains that have occasionally been associated with human disease, but also includes a large number of strains that primarily serve as nitrogen-fixing bacterial symbionts in legumes (and it has been cited as a common laboratory contaminant by Salter et al., 2014).

o Lines 41-42: Is it concerning that Proteobacteria were identified as the most abundant phylum in the sample set, at nearly 65%? In published reports of small mammal spleen microbiomes, Firmicutes have been the most relatively abundant and Proteobacteria have accounted for a smaller fraction (e.g., Ge et al., 2018).

o Line 52: “the study confirms the role of the spleen as a microbial filter”—the study did not report on the function of the spleen as a filter, but instead used prior knowledge that the spleen has this function to investigate the bacterial 16S rRNA sequences that were present in the tissue.

Additional comments

I was not assigned to review the original manuscript and this I evaluated the manuscript in light of the most recent version.

I was quite pleased to see the combination of host DNA barcoding and bacterial profiling in this paper. I was especially excited to learn more about the bacterial profiles obtained from spleens in wild small mammals, and appreciated the speculation about how these profiles might be connected to variation in local land uses (e.g., cattle ranching, farming, etc.).

---

## Round 0.3 · accepted · Accept

Thank you for addressing the reviewers' comments adequately and in detail.